# Fast Training of Contrastive Learning with Intermediate Contrastive Loss

## Abstract

Recently, representations learned by self-supervised approaches have significantly reduced the gap with their supervised counterparts in many different computer vision tasks. However, these self-supervised methods are computationally challenging. In this work, we focus on accelerating contrastive learning algorithms with little or even no loss of accuracy. Our insight is that, contrastive learning concentrates on optimizing similarity (dissimilarity) between pairs of inputs, and the similarity on the intermediate layers is a good surrogate of the final similarity. We exploit our observation by introducing additional intermediate contrastive losses. In this way, we can truncate the back-propagation and updates only a part of the parameters for each gradient descent update. Additionally, we do selection based on the intermediate losses to filter easy regions for each image, which further reduces the computational cost. We apply our method to recently-proposed MOCO (He et al., 2020), SimCLR (Chen et al., 2020a), SwAV (Caron et al., 2020) and notice that we can reduce the computational cost with little loss on the performance of ImageNet linear classification and other downstream tasks.

## 1 Introduction

Recently, self-supervised learning has been shown a promising approach for unsupervised and semi-supervised learning in computer vision (Oord et al., 2018; Kolesnikov et al., 2019; Zhai et al., 2019; He et al., 2020; Chen et al., 2020b;a; Grill et al., 2020). These methods learn unsupervised representation that can perform well on both ImageNet (Deng et al., 2009) linear classification and other down-stream tasks, e.g. pose estimation, detection, and semantic segmentation (He et al., 2020; Caron et al., 2020). A major type of self-supervised learning is contrastive learning, which constructs similar and dissimilar pairs over the dataset and minimize a constrative loss to learn a mapping that yields similar (resp. dissimilar) pairs with similar (resp. dissimilar) representations.

Despite the recent successes, contrastive learning was found to practically incur longer training time and higher computational cost compared with supervised learning (He et al., 2020; Grill et al., 2020). For example, He et al. (2020) and Chen et al. (2020a) requires $5 \times$ time than standard supervised learning on ImageNet. The enormous time and computational cost makes large-scale contrastive learning out of reach for many of the researchers and applications.

This work focuses on speed up contrastive learning. Our key observation is, because contrastive learning focuses on optimizing similarity (dissimilarity) between pairs of inputs, the similarity on the intermediate layers provide a good surrogate of the final similarity, and computing the intermediate representation requires less computational cost. This is in contrast with supervised learning, which requires to match the output of the final layer with a label and hence it is essential to calculate the final outputs.

We exploit the observation by introducing additional contrastive losses in the middle layers of the neural network. Instead of measuring only the contrastive loss of the representation in the last layer, we compute the contrastive loss in the intermediate blocks. The intermediate losses enable the following two strategies that accelerates contrastive learning. (1) *Partial Back-propagation*: We start back-propagation randomly from one of all the contrastive losses. Compared with doing full back-propagation in every optimization step, an intermediate starting point only requires computing the gradients for a part of the parameters. (2) *Block-wise Hard Pair Selection*: The intermediate

contrastive losses can serve as indicators of the similarity between the representations in early layers. These indicators can be used to filter out the simple pairs, thus reducing unnecessary computation.

We test the proposed method upon several recent self-supervised learning algorithms, MOCO (He et al., 2020), SwAV (Caron et al., 2020), simCLR(Chen et al., 2020a) and MOCO V2 (Chen et al., 2020b). We empirically show that our method can save the training time with almost no loss on the final performance of the downstream tasks, e.g. ImageNet linear classification, PASCAL VOC object detection and segmentation. Our method largely reduces the training cost for contrastive learning methods by over 30%, and can serve as an alternative to standard self-supervised learning training pipeline if the computation resources is limited.

## 2 METHOD

We first give a brief introduction about contrastive learning in Sec. 2.1, and then introduce our method in Sec. 2.2 and Sec. 2.3. Our method is composed of two major components: (1) using randomly early stopping for different mini-batch (2) using random crop and selecting hard regions on hidden states.

### 2.1 CONTRASTIVE LEARNING

Given an unlabeled set of images $\mathcal{D} = \{x_i\}$, we want to learn a representation map $f$ that extracts useful low-dimensional representations from the high-dimensional images $x$. In contrastive learning, for each $x \sim \mathcal{D}$, we use data augmentation or other techniques to construct a positive example $x_+$ that is similar $x$ and a set of negative examples $\{x_{-k}\}_{k=1}^{K}$ that are less similar to $x$ than $x_+$. Then we train the map $f$ to maximize the similarity between $f(x)$ and $f(x_+)$, while minimizing the similarity between $f(x)$ and $f(x_{-k})$. A popular choice of contrastive loss is InfoNCE (Oord et al., 2018),

$$\mathcal{L}_{\text{info}}(f(x), f(x_+)) = -\log \frac{\exp\left(\frac{1}{\tau} f(x)^\top f(x_+)\right)}{\sum_{k=1}^{K} \exp\left(\frac{1}{\tau} f(x)^\top f(x_{-k})\right) + \exp\left(\frac{1}{\tau} f(x)^\top f(x_+)\right)} \qquad (1)$$

The encoder aims to minimize the InfoNCE for all the images in the dataset,

$$\mathcal{L}(f) = \mathbb{E}_{x \sim \mathcal{D}} \left[ \mathcal{L}_{\text{info}}(f(x), f(x_+)) \right], \qquad (2)$$

where $\tau$ is a temperature hyper-parameter, Intuitively, Eq. 2 is the log loss of a $(K+1)$-way softmax-based classifier that tries to classify $f(x)$ into the same class as $f(x_+)$.

There are many different methods to construct the positive and negative examples. In practice, people use context (Oord et al., 2018), data augmentation (Chen et al., 2020a; He et al., 2020; Hénaff et al., 2019), colorization (Oord et al., 2018), clustering (Caron et al., 2020; 2018), etc., to construct the positive pair $\{x, x_+\}$. More generally, each image $x$ can have multiple positive examples (Tian et al., 2019). Previous works have used in-batch data (Chen et al., 2020a), memory bank (He et al., 2020), or different regions of one given image (Oord et al., 2018) to generate $\{x_{-k}\}_{k=1}^{K}$ for each $x$.

### 2.2 ACCELERATED TRAINING BY PARTIAL BACK-PROPAGATION

Our approach stems from a fundamental property of contrastive representation learning: contrastive loss measures the similarity between representations. Due to the hierarchical nature of deep neural networks, similar representation pairs from early layers are still similar even after being processed by the later layers. Therefore, we argue that full back-propagation is not necessary for contrastive representation learning. Rather, partial back-propagation is enough for learning useful representations in the final layer. Intermediate loss has been widely used in deep learning, from Inception Network (Szegedy et al., 2015) to DARTs (Liu et al., 2019).

We consider a deep neural network $f = f_n \circ f_{n-1} \circ \cdots f_1$, where $f_i$ refers to the $i$-th building block of the network. For feed-forward neural networks, the building block can be one hidden layer. For more complex network, e.g. ResNet (He et al., 2016), the building block can be a sequential of convolutional layer, batch normalization layers (Ioffe & Szegedy, 2015), and activation functions.

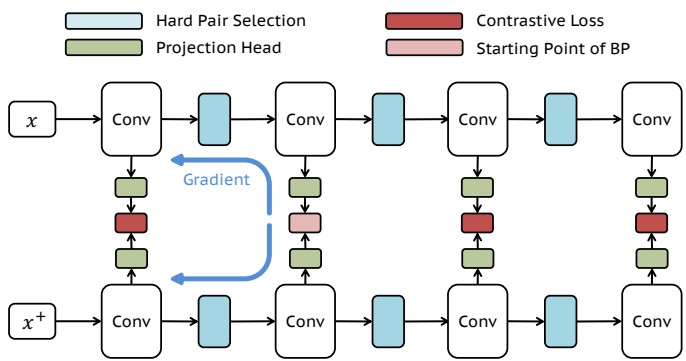

Figure 1: Demonstration of our partial back-propagation framework during training. The network in the figure has 4 blocks. For each block, we introduce additional projection head and contrastive loss. In this optimization step, we randomly choose to start back-propagation from the contrastive loss of the second block.

---

**Algorithm 1** Accelerated Training by Partial Back-Propagation

**Input**: A network $f = f_n \circ f_{n-1} \circ \cdots f_1$, a dataset $\mathcal{D}$.
**repeat**
    Randomly sample a mini-batch $\{x_k\}_{k=1}^m$ from $\mathcal{D}$.
    Randomly sample $\mathcal{L}_i$ from $S_{loss}$.
    Update $f_j, j \in \{1, ..., i\}$ by mini-batch gradient descent on the loss $\frac{1}{m} \sum_{k=1}^m \mathcal{L}_i$.
**until** Convergence

---

Instead of using a single contrastive loss $\mathcal{L}$ at the end of network, we propose to add auxiliary contrastive losses $\mathcal{L}_i$ after each building block $f_i$, obtaining a set of losses $S_{loss} = \{\mathcal{L}_1, \ldots, \mathcal{L}_n\}$. Expanding $\mathcal{L}_i$ yields,

$$\mathcal{L}_i = \mathcal{L}(f_{1:i}) = \mathbb{E}_{x \sim \mathcal{D}} \left[ \mathcal{L}_{\text{info}}(f_{1:i}(x), f_{1:i}(x_+)) \right] \tag{3}$$

Here, $f_{1:i}(x)$ is the intermediate representation after block $f_i$, $f_{1:i}(x) = f_i \circ \cdots \circ f_1(x)$. Note that we ignore the auxiliary projection head here for simple notation. In every optimization step, we randomly choose a loss $\mathcal{L}_i$ from $S_{loss}$, and start back-propagation from $\mathcal{L}_i$.

Because back-propagation from $\mathcal{L}_i$ does not involve blocks $f_{i+1}$ to $f_n$, our method can reduce the computational burden from two aspects. (1) Reduction in the number of Multiply-Accumulation operations (MACs). For example, consider the backward pass of a standard ResNet-50 for one iteration, in the average case, our method can reducing the 8.2G MACs (He et al., 2016) to 3.6G MACs. (2) Reduction in the storage and other additional costs. When training a convolutional network, a great proportion of the cost is the layer-wise storage and callback of the intermediate outputs and gradients (Chen et al., 2016; Yu et al., 2014; Kusumoto et al., 2019). Our method decreases the depth of the back-propagation chain, and consequently decreases the additional storage and callback cost. Overall, both aspects ensure our method to speed up the backward process in training.

## 2.3 BLOCK-WISE HARD PAIR SELECTION

Besides reducing the burden in back-propagation, the auxiliary losses enable us to judge whether the representations of $x$ and $x_+$ are similar in early stages of the forward pass. We can filter out the simple pairs to further reduce the size of the computational graph, and in the meanwhile render the network focus on the difficult pairs. Our strategy not only significantly decreases the training time, but also helps recover the performance of standard training process, as demonstrated in Sec. 4.

For $x$ and $x_+$, let their intermediate representations after the $i$-th block be $f_{1:i}(x), f_{1:i}(x_+) \in \mathbb{R}^{C \times H \times W}$, where $C, H, W$ denote the number of channels, height, and width. We crop $f_{1:i}(x)$ and $f_{1:i}(x_+)$ to obtain smaller patches and select the difficult pairs from them. We apply two-dimensional random crop on $f_{1:i}(x)$ and $f_{1:i}(x_+)$ for $M$ times, yielding two sets of smaller patches, $B = \{f_{1:i}^1(x), \ldots, f_{1:i}^M(x)\}$ and $B^+ = \{f_{1:i}^1(x_+), \ldots, f_{1:i}^M(x_+)\}$. Here, $f_{1:i}^j(x), f_{1:i}^j(x_+) \in$

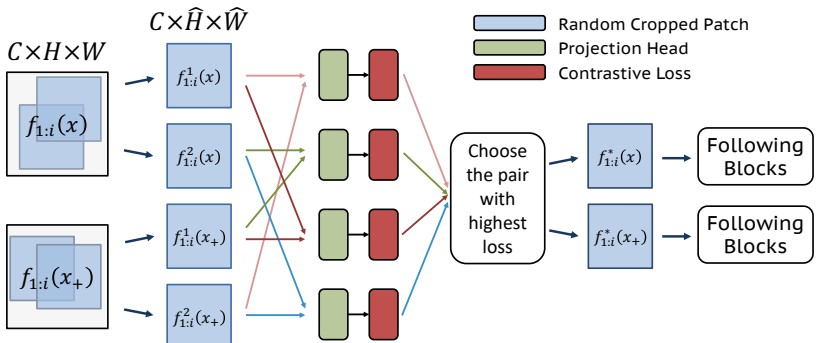

Figure 2: Demonstration of hard pair selection when $M = 2$. From all the four pairs, the pair with the highest contrastive loss is selected and passed to the following blocks. The procedure introduces little computational overhead except for a forward pass through the projection head. In comparison, a smaller feature map significantly decreases the computational cost of the whole network.

$\mathbb{R}^{C \times \hat{H} \times \hat{W}}$ are cropped patches, and $\hat{H} < H, \hat{W} < W$. We select the hard pair as follows,

$$f_{1:i}^*(x), f_{1:i}^*(x_+) = \underset{f_{1:i}^p(x) \in B, f_{1:i}^q(x_+) \in B^+}{\arg\max} \mathcal{L}_{\text{info}}(f_{1:i}^p(x), f_{1:i}^q(x_+)), \quad p, q \in \{1, 2, \ldots, M\}. \quad (4)$$

The above criterion picks the pair of representations with the largest contrastive loss. We then feed only $f_{1:i}^*(x)$ and $f_{1:i}^*(x_+)$ to the later blocks in the network. Here, we only process the positive pairs, and do nothing for the negative pairs. Taking the properties of convolutional neural networks into account, our method can be viewed as progressively reducing the resolution of the original input images. We adopt standard image resolution for the first block, and apply our strategy in the others.

As noted in prior works (e.g., Chen et al., 2020a; Misra & Maaten, 2020), applying random crops on an image plays an essential role in self-supervised learning. Caron et al. (2020) uses multiple crops of given images to do clustering, outpeforming standard SimCLR (Chen et al., 2020a) and other baselines (e.g. He et al., 2020; Chen et al., 2020b; Hénaff et al., 2019). People also use multiple random crops and select the hard examples (Gong et al., 2020) to boost the performance. However, selecting the hard augmentation examples or using multi-crop will increase the computational cost, because they require multiple forward passes of the whole network for multiple input images. Our method, in contrast, acts on the feature map and filters the simple pairs in every intermediate block, resulting in both performance improvement and reduction of training time.

## 3 RELATED WORKS

Training a deep neural network usually requires high computation costs. Therefore, a long line of works have been devoted to accelerating the training process of deep neural networks. Unlike efficient inference, methods for efficient training have been developed case by case (Han et al., 2016b;a; Ye et al., 2020) Typically, researchers use domain knowledge to design different strategies for different tasks to accelerate the training (Wu et al., 2020; Singh et al., 2018). In this work, we also leverage the special property of contrastive learning to develop the method.

**Runtime Pruning** Runtime Pruning removes redundant channels during training and uses the whole network with all removed channels for inference Lin et al. (2017) uses an RNN controller to remove useless filters during training. the controller decides if a channel should be removed based on the feature map as inputs. Gao et al. (2018) replaces the RNN controller with another neural network The runtime pruning method can be applied to many of the supervised learning tasks. However, using the controller will introduce additional computation cost (Wang et al., 2019), and therefore usually could not accelerate the training by a large margin in practice. Moreover, the tuning of the controller is still an ongoing problem.

**Early skipping** Early skipping techniques reduce computation by skipping part of the neural network whenever it is unnecessary For example, in Figurnov et al. (2017), an adaptive number of ResNet layers are skipped within a residual block for unimportant regions in object classification.

The skipping mechanism is controlled by a halting scores predicted at the output branch of each residual unit. Li et al. (2017) handles easy regions in the early stages and harder cases are progressively fed forward to the next sub-model for further processing. These approaches are usually useful for specific tasks, e.g. segmentation, detection, which can take benefits from low-level features. For more general tasks, e.g. classification, it fails to achieve good performance (Li et al., 2017).

**Multi-scale training** Multi-scale training is a general approach in computer vision community, e.g. segmentation (He et al., 2017), detection (Singh et al., 2018), video (Wu et al., 2020). However, the inevitable challenge is that the strategy is task-dependent or dataset-dependent. However, finding a general dynamic schedule for different cases to adjust resolution is extremely difficult (Wu et al., 2020; Simonyan & Zisserman, 2014). Usually, the strategy is a bit complicated. For example, Wu et al. (2020) proposes to schedule the batch size with both long and short cycles.

**Summary and Discussion** All these works are mainly based on two techniques: reducing resolution and skipping layers. Most of these methods are developed by either hand design or learning-based method, e.g. reinforcement learning (Lin et al., 2017). Compared to hand-designed method, the learning-based method introduces more hyper-parameters (which is not easy to tune) while the design of search space still highly dependents on human knowledge. Therefore, in practice, it usually cannot find solution as good as hand-crafted method or grid search. In this work, we design the acceleration method without the help of learning-based approach.

## 4 EXPERIMENTS

In experiments, we verify whether our proposed method can accelerate recent self-supervised learning approaches with no loss on the performance on different tasks. We choose MOCO (He et al., 2020), simCLR (Chen et al., 2020a), SwAV (Caron et al., 2020) and MOCO V2 (Chen et al., 2020b) as our benchmark, and test the performance on object detection, segmentation and classification. Our results are averaged over 5 trials for all the experiments.

**Training Settings** We train the self-supervised methods on ImageNet (Deng et al., 2009) training set that has 1.28 million images in 1000 classes. This dataset is well-balanced in its class distribution, and is the most commonly-used benchmark to train self-supervised representations.

We implement all the baseline algorithms and our methods all on the most widely-used ResNet-50 (He et al., 2016). For the hyper-parameters used in the training process, we keep almost all the hyper-

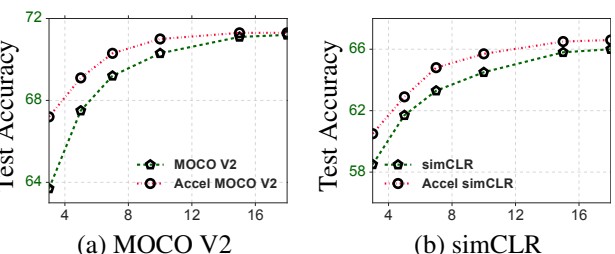

Figure 3: Linear classification on ImageNet. The x-axis shows the training time (days), and the y-axis shows the top-1 accuracy for the linear classifier.

parameters the same as their reporting in the paper, except batch size and learning rate. As our method can save GPU memory by selecting hard pair, we can use a larger batch size. In practice, we set the batch size as large as possible and linearly scale the learning rate (He et al., 2019). We train all the methods and models on 8 GPUs [1] and compare with all the baselines trained in the same settings and machines.

For the layer-wise loss on ResNet-50, we set three auxiliary head (loss) after the three blocks. The projection head is several linear transformations followed by a softmax layer and the contrastive loss (Chen et al., 2020a). For the hard region pair selection, we select hard pairs after the first and second block. The size of the selected regions is 3/4 of the original size.

**Downstream Settings** After training the self-supervised representations, we verify the performance on several downstream tasks. The main goal of unsupervised learning is to learn transferrable features, therefore we test the performance on different tasks except for the simple linear classification. We first verify the performance on ImageNet linear classification task with frozen features, which

---

[1] We use NVIDIA TITAN V VOLTA 12GB, and record the training time for this setting.

is the most used benchmark. Then, we verify the performance on segmentation and detection tasks, with both frozen features and unfrozen features.

## 4.1 IMAGENET LINEAR CLASSIFICATION

We evaluate the representations trained with various methods on ImageNet. We train all the models with 200, 400 or 800 epochs. For SwAV, we do not apply the multi-crop strategy described in their paper. We run the linear classification three times and report the final results.

We show the primary result in Table 1. It shows that applying to different self-supervised learning algorithms and different epochs, our method can always accelerate the training with little loss of accuracy. For simCLR and SwAV, we can even boost the top-1 accuracy (e.g. from 61.7% to 62.9% for simCLR and from 70.1% to 70.7% for SwAV) with less training time. We further show the relationship between training time and top-1 accuracy in Figure 3. With the same time budget, the proposed algorithm can improve the performance of the linear classifier. Especially when the time budget is limited (e.g. 3 days on 8 GPUs), the proposed accelerating training method can boost the performance with a large margin. For example, as shown in Figure 3, we can improve the MOCO V2 performance from 64% to 66.5% when the training time is limited to 3 days. When more computation resources are given, our proposed method will converge to the same performance as the standard version.

| Method | Epoch | Top-1 | Time (Days) |
|---|---|---|---|
| simCLR | 200 | 61.7±0.2 | 4.3 |
| Accel-simCLR | 200 | 62.9±0.2 | **2.8** |
| MOCO | 200 | 60.6±0.2 | 4.3 |
| Accel-MOCO | 200 | 60.5±0.1 | **2.8** |
| MOCO V2 | 200 | 67.5±0.1 | 4.3 |
| Accel-MOCO V2 | 200 | 67.3±0.1 | **2.8** |
| MOCO V2 | 800 | 71.1±0.1 | 17.2 |
| Accel-MOCO V2 | 800 | 71.0±0.2 | **10.9** |
| SwAV | 400 | 70.1±0.1 | 8.1 |
| Accel-SwAV | 400 | 70.7±0.1 | **5.3** |

Table 1: Linear classification on ImageNet. Top-1 accuracy for linear models trained on frozen features from different self-supervised methods.

## 4.2 TRANSFERRING FEATURES

Since the main propose for self-supervised learning is to get a universal representation, it is necessary to check whether the learned representations can be transferred to more downstream tasks. We compare our accelerating method and the standard versions, transferred to various tasks including PASCAL VOC 2007 and PASCAL VOC 2012. Followed MOCO (He et al., 2020), we use an additional BN layer after the backbone, and synchronize the BN statistics across GPUs. We test the following models: simCLR trained with 200 epochs, MOCO trained with 200 epochs, MOCO V2 trained with 800 epochs and SwAV trained with 400 epochs.

**VOC 2012 Segmentation** We first evaluate all the models on PASCAL VOC 2012 segmentation tasks, and show the results in Table 2. We test two different settings: freezing the backbone or not, and report two metrics: mean IoU (Intersection-Over-Union) and pixel-wise accuracy. We use PSP-Net (Zhao et al., 2017) as the decoder. Table 2 shows that for all these settings and different models, our proposed accelerating method would not drop the performance on VOC 2012 segmentation task. Because of the small standard deviation (e.g. $< 0.1$), we do not list them in the table. We notice that although our accelerated MOCO V2 performs worse than standard MOCO V2 on ImageNet linear classification, they achieve the same performance on this task. On the other hand, the accelerated SwAV is no longer better than standard SwAV on this task. It indicates that we should test the algorithm on various tasks to verify whether it would cause a drop in the performance.

**VOC 2007 Object Detection** We further test all these models on the VOC 2007 object detection task. The detector is Faster R-CNN with R50-C4 (He et al., 2017), using `Detectron2` (Wu et al., 2019). The image scale is [480, 800] pixels during training and [800, 800] at inference. We evaluate

| Method | Freeze | mean IoU | Accuracy | Freeze | mean IoU | Accuracy |
|---|---|---|---|---|---|---|
| Supervised | √ | 76.9±0.1 | 85.0±0.0 | × | 79.1±0.1 | 87.1±0.2 |
| simCLR | √ | 72.4±0.1 | 82.0±0.0 | × | 74.8±0.1 | 83.4±0.1 |
| Accel-simCLR | √ | **72.9±0.0** | 82.3±0.1 | × | **75.2±0.1** | 83.7±0.1 |
| MOCO | √ | **75.1±0.1** | 83.2±0.1 | × | **77.2±0.1** | 84.8±0.1 |
| Accel-MOCO | √ | **75.2±0.2** | 83.3±0.1 | × | 76.9±0.1 | 84.6±0.1 |
| MOCO V2 | √ | **78.6±0.2** | 86.6±0.2 | × | **79.4±0.2** | 87.3±0.2 |
| Accel-MOCO V2 | √ | **78.6±0.2** | 86.5±0.2 | × | 79.3±0.2 | 87.3±0.2 |
| SwAV | √ | **72.7±0.1** | 80.7±0.0 | × | 74.4±0.2 | 83.5±0.1 |
| Accel-SwAV | √ | **72.7±0.1** | 80.8±0.0 | × | **74.5±0.2** | 83.5±0.1 |

Table 2: Image segmentation on PASCAL VOC 2012. The data is per-processed with (Hariharan et al., 2011). For 'freeze', √ denotes we freeze the backbone parameters, while × denotes the parameters is not frozen. We report mean IoU (Intersection-Over-Union) and pixel-wise accuracy to measure the performance.

the default VOC metric of AP50 (i.e., IoU threshold is 50%) and the more widely-used metrics, COCO-style AP. We do training on `VOCtrainval2007` and do valuation on the `VOCtest2007`. We display the experiment results in Table 3. On VOC 2007 object detection tasks, our accelerated versions do not have worse performance compared to the standard version for all the tested methods.

| Method | Freeze | AP50 | AP | Freeze | AP50 | AP |
|---|---|---|---|---|---|---|
| Supervised | √ | 72.8±0.0 | 39.5±0.1 | × | 74.4±0.0 | 42.4±0.1 |
| simCLR | √ | 71.2±0.0 | 37.4±0.2 | × | 73.3±0.1 | 40.1±0.2 |
| Accel-simCLR | √ | 71.4±0.0 | **37.8±0.1** | × | 73.6±0.0 | **40.6±0.1** |
| MOCO | √ | 73.6±0.1 | **40.5±0.1** | × | 77.9±0.0 | 46.6±0.1 |
| Accel-MOCO | √ | 73.6±0.0 | **40.5±0.1** | × | 78.0±0.1 | **46.7±0.1** |
| MOCO V2 | √ | 77.7±0.1 | **46.5±0.2** | × | 80.2±0.0 | **50.5±0.1** |
| Accel-MOCO V2 | √ | 77.5±0.1 | 46.2±0.1 | × | 80.1±0.0 | 50.2±0.1 |
| SwAV | √ | 74.2±0.1 | 42.1±0.2 | × | 78.2±0.0 | 46.8±0.1 |
| Accel-SwAV | √ | 74.3±0.1 | **42.3±0.2** | × | 78.4±0.1 | **46.9±0.1** |

Table 3: Object detection on PASCAL VOC 2007. For 'freeze', √ denotes freezing the backbone parameters, while × denotes the parameters is not frozen. We report AP (average precision) and AP50 to measure the performance.

### 4.3 FURTHER STUDY

We further do some study to empirically understand how each part of the proposed accelerating method contributes to the final result.

**Layers Close to the Input are Important** We first study which part of a given neural network is important or even necessary for the representation learning during training.

Shown in Table 4, we test which layer could be skipped during training. We test three different settings: 1) randomly selecting some not adjacent layers to do backward, and pass the information through the not selected layers with skip connection 2) stop-

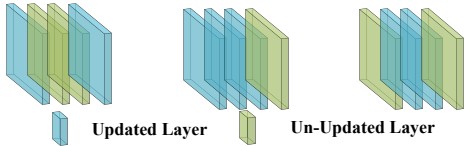

Figure 4: The three update scheme described in context.

ping forward at one certain layer, and then doing backward to the input 3) selecting several nearby blocks to do backward and not do always backward to input. We display these three approaches in Figure 4. We test these methods on MOCO V2 and simCLR, and report the results in Table 4. Table 4 displays that doing backward to input achieves best performance, whle the other two approaches achieve worse results. It indicates that for these self-supervised algorithms, the blocks close to the input need more updates and gradient information. Recent work (Zhang et al., 2019) also shows that for supervised image classification, the layers close to the input are important for many different neural architectures.

| Method | Accuracy | Time (GPU Days) | Method | Accuracy | Time (GPU Days) |
|---|---|---|---|---|---|
| MOCO V2 | 67.5±0.1 | 4.3 | simCLR | 61.7±0.2 | 4.3 |
| Skipping | 65.3±0.2 | 2.8 | Skipping | 61.5±0.3 | 2.8 |
| BP-to-Input | **67.3±0.1** | 2.8 | BP-to-Input | **62.9±0.2** | 2.8 |
| Nearby-Block | 65.5±0.2 | 2.8 | Nearby-Block | 61.3±0.2 | 2.8 |

Table 4: Tested on ImageNet linear classification task, we empirically study which layer is important for self-supervised learning.

**Hard Pairs v.s. Easy Pairs v.s. Hard Examples** In our framework, we filter easy regions (i.e. regions with lower contrastive loss) in the hidden states. To fully understand the role of the operation, we compare to two baselines under the same computation budget: 1) select easy region, 2) select hard examples in each mini-batch.

We show the performance of all these three approaches in Table 5. It demonstrates that selecting easy pairs drops the performance (e.g. for MOCO V2, compared to selecting hard pairs, selecting easy pairs will break top-1 accuracy from 67.3% to 65.3%). We also notice that selecting hard examples would be a bit worse than selecting hard region pairs. This is reasonable for contrastive learning since many researchers have found that using multi-crop is helpful for contrastive learning, and can even boost the performance in downstream tasks in some cases (Grill et al., 2020; Chen et al., 2020a; Misra & Maaten, 2020). Moreover, selecting hard region pairs seize out the meaningful hard context for each given image. Predicting meaningful context itself can be useful for self-supervised learning (Oord et al., 2018).

| Method | Accuracy | Method | Accuracy |
|---|---|---|---|
| MOCO V2 | 67.5±0.1 | simCLR | 61.7±0.2 |
| Hard Pair | **67.3±0.1** | Hard Pair | **62.9±0.2** |
| Easy Pair | 65.3±0.3 | Easy Pair | 60.3±0.4 |
| Hard Example | 66.1±0.2 | Hard Example | 61.6±0.2 |

Table 5: We demonstrate an study of the role of selecting hard region pairs. Compared to selecting hard examples or easy regions, selecting hard region pairs can achieve better performance.

**Ablation Study** Finally, we conduct an ablation study on simCLR and MOCO V2 to see the advantages of each part for our proposed algorithm. As displayed in Table 6, for MOCO V2, both the partial back-propagation and the block-wise hard pair selection can achieve similar accuracy as the standard version while using fewer computation resources. For simCLR, hard pair selection can even boost the performance of the original version with less training time, while partial back-propagation achieves similar performance as the standard simCLR. Combining these two approaches together can further save training time while having no loss of accuracy for both MOCO V2 and simCLR.

| Method | Accuracy | Time (GPU Days) | Method | Accuracy | Time (GPU Days) |
|---|---|---|---|---|---|
| MOCO V2 | 67.5±0.1 | 4.3 | simCLR | 61.7±0.2 | 4.3 |
| + PB | 67.3±0.2 | 3.7 | + PB | 61.8±0.1 | 3.7 |
| + HP | 67.1±0.1 | 3.4 | + HP | **63.1±0.2** | 3.4 |
| + PB + HP | **67.3±0.1** | **2.8** | + PB + HP | 62.9±0.2 | **2.8** |

Table 6: 'PB' denotes partial back-propagation, 'HP' denotes hard pair selection for each image pair.

## 5    CONCLUSION AND DISCUSSION

Recent self-supervised learning has achieved great success in computer vision, e.g., matching or even outperforming the supervised learning model on some vision tasks. However, the huge computation cost prevents many researchers from using it on their research topics. In this paper, we focus on accelerating the training of self-supervised learning based on contrastive loss. Empirically, we show that our proposed method can achieve the same performance as several recent proposed self-supervised learning methods, with far less training time. In the future, we plan to further study

the theoretical property of these self-supervised approaches, to see whether it has some inherent property to achieve better performance or faster convergence. Furthermore, we will test our method on more tasks to see whether it is a more general approach.

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

## A  IMPLEMENT DETAILS

For the training of self-supervised learning algorithm, we use the open-source platforms [2] to implement and keep almost all the hyper-parameters the same, except learning rate and batch size. For simCLR, we use 256 batch size, which achieves sightly worse result than Chen et al. (2020a) reported. For SwAV which use cluster centers as positive examples, we keep additional block-wise cluster centers for the loss. For MOCO and MOCO V2 which use memory bank, we keep additional layer-wise queen (memory bank), which costs additional memory but does almost no hurt to training time.

For detection and segmentation, we use open-source code base [3] and keep the hyper-parameters the same. We add an additional BN layer after the backbone for this setting.

## B  ABLATION STUDY

In the section, we do more ablation study to show the impact of different parts of our approach. We mainly show the result on simCLR and SWAV.

### B.1  DIFFERENT NUMBERS OF SELECTED REGION

In Eq (4), we select hard pair over $M$ candidates. In practice, we notice that for simCLR trained with 200 epochs, the number of candidates $M$ does not impact the final performance a lot. We show the results in Table 7.

| $M$ | 1 | 2 | 4 | 10 |
|---|---|---|---|---|
| Accuracy | 62.4±0.2 | 62.8±0.1 | 62.9±0.2 | 62.9±0.2 |

Table 7: Top-1 accuracy for linear models trained on frozen features on ImageNet. We change the number of candidates for selecting hard pairs in this table.

### B.2  INTERMEDIATE REPRESENTATIONS

We use the intermediate representations of simCLR and Accel-simCLR to do the linear classification task for ImageNet. As shown in Table 8, the intermediate representations are improved when intermediate loss is used. Compared to the final layer representations, the intermediate representations of simCLR and Accel-simCLR have a larger gap on ImageNet top-1 accuracy.

---

[2]https://github.com/open-mmlab/OpenSelfSup, https://github.com/facebookresearch/moco, https://github.com/facebookresearch/swav

[3]Detectron2 and https://github.com/yassouali/pytorch_segmentation

|  | Feat1 | Feat2 | Feat3 | Feat4 | Final |
|---|---|---|---|---|---|
| simCLR | 17.6±0.2 | 31.6±0.2 | 41.9±0.2 | 54.1±0.2 | 61.7±0.2 |
| Accel-simCLR | 19.4±0.2 | 33.5±0.3 | 43.5±0.2 | 55.6±0.2 | 62.9±0.2 |

Table 8: Top-1 accuracy for linear models trained on frozen features on ImageNet. 'Feat n' represents the hidden after $n$-th block in ResNet-50, and 'Final' denotes the final-layer representations.

| Method | Epoch | Top-1 | Time (Days) |
|---|---|---|---|
| $2 \times 224$ | 400 | 70.1±0.1 | 8.1 |
| Accel-$2 \times 224$ | 400 | 70.7±0.1 | **5.3** |
| $2 \times 160 + 4 \times 96$ | 400 | 74.2±0.1 | 8.8 |
| Accel-$2 \times 160 + 4 \times 96$ | 400 | 74.1±0.1 | **5.7** |

Table 9: We conduct different multi-crop settings for SwAV.

### B.3 MULTI-CROP FOR SWAV

For SwAV, we test two different settings, using $2 \times 224$ and $2 \times 160 + 4 \times 96$ during training for a given image. $2 \times 224$ denotes for one image, we generate two $224 \times 224$ augmented images. $2 \times 160 + 4 \times 96$ denotes for one image, we generate two $160 \times 160$ and four $96 \times 96$ augmented images. Table 9 denotes that, for both settings, we can accelerate the original version without loss on accuracy.

| resolution | 224 | 196 | 128 |
|---|---|---|---|
| Accuracy | 67.3±0.1 | 60.6±0.2 | 53.5±0.2 |

Table 10: Top-1 accuracy for linear models trained on frozen features on ImageNet. We change the resolution for selecting hard pairs in this table. We do the experiments on MOCO V2, using the official implementation[5].

### B.4 REDUCING RESOLUTION HURTS PERFORMANCE

A straightforward way to accelerate the training process is to use smaller image resolution. Shown in Table 10, for MOCO V2, reducing the image resolution hurts the performance of the ImageNet linear classification by a large margin. For example, using $128 \times 128$ resolution only achieves around $53.5\%$ top-1 accuracy on ImageNet.

