# OpenReview forum: "Fast Training of Contrastive Learning with Intermediate Contrastive Loss"
_ICLR.cc/2021/Conference — Reject_

### Official Review · AnonReviewer4 · 2020-10-26
**Slight acceleration achieved, but much more to be explored?**

**Rating:** 6
**Confidence:** 4

**Review:**

In this submission the authors suggest modifications to reduce the computational cost of contrastive learning. The authors propose to add constrastive loss heads at intermediate stages of the network. Using these intermediate losses as a proxy for the final contrastive loss they make two proposals for  reducing computational effort: 1) randomly choosing which contrastive loss to start from such that on average less of the network is actually applied. 2) randomly cropping the image at each intermediate loss and continuing only with hard patches for the positive samples.

Overall, I am on the fence for this one. I think adding the intermediate contrastive losses is an interesting idea and the proposed changes already improve training times. However, the reductions in training times are not overwhelming and I believe there would be many more aspects to explore for earlier contrastive losses. Thus, I am not overly enthusiastic.

Pros:
1) Earlier contrastive losses are an interesting idea!
2) There is a reduction in training times at negligible cost in performance.
3) The method is quite general over contrastive learning methods.

Cons:
1) The reduction in training times is not overwhelming given the amount of changes.
2) While the authors present a few analyses which parts of their method and network are important I don’t think these are particularly informative.
3) Having losses for the individual parts of the network, the immediate idea would be to train the parts separately which would remove many more backpropagation operations and would allow parallelization of the network parts onto separate machines. This possibility is not explored here.
4) Given that the stated aim is to reduce training times it seems that there is very little analysis on what actually takes time in this approach.

---

> ### Author Response · Authors · 2020-11-18
> **Reponse to Questions**
>
> We thank reviewer#4 for the thoughtful feedback!
>
> “The reduction in training times is not overwhelming given the amount of changes.”
>
> -- Our approach can yield a speedup of 1.2x to 1.5x with no loss on the accuracy on several downstream tasks. This is in fact a large-margin improvement compared with acceleration methods for supervised learning such as runtime pruning methods (lin et. al 2017, gao et. al 2018).
> Importantly, we are the first work to speed up contrastive learning to the best of our knowledge. We believe that (also as recognized by other reviewers), our basic framework opens the door to various further acceleration that can be explored in future works,  e.g., by using large batch size (e.g. 65536), more machines (e.g. 256 GPUs), new optimizers.
>
> “Given that the stated aim is to reduce training times it seems that there is very little analysis on what actually takes time in this approach.”
>
> -- As shown in Table 6, partial BP can save ~18% time and hard pair mining can further save ~20% time. In our approach, backpropagation still causes most of the time.
>
> “While the authors present a few analyses which parts of their method and network are important I don’t think these are particularly informative.”
>
> -- We studied the impact of different parts in the experiments. In the revision, we will conduct more ablation studies to understand the importance of each part/design of our method. We would highly appreciate it if the reviewer could provide suggestions on how to make the analysis more informative.
>
> “Having losses for the individual parts of the network, the immediate idea would be to train the parts separately which would remove many more backpropagation operations and would allow parallelization of the network parts onto separate machines. This possibility is not explored here.”
>
> -- We have done this ablation. This can be viewed as a special case of Figure 4(a). If we separate all the blocks and do BP, the performance of MOCO V2 is not comparable to other results in Table 4. The approach we proposed strikes a balance of cost and accuracy between fully separate layerwise training and the traditional BP.

---

### Official Review · AnonReviewer1 · 2020-10-28
**Promising initial results, curious to see if/how it scales to more pre-training epochs**

**Rating:** 6
**Confidence:** 4

**Review:**

This paper proposes techniques to speedup contrastive self-supervised pre-training. The two key ideas are to back-prop only through a subset of layers in the network (from a random layer, back to the input), and to drop training instances which are not "hard" (in a way defined more precisely in the paper). By combining these techniques the paper shows that pre-training time can be reduced by ~30% without significant loss in performance on downstream tasks.

The proposed acceleration techniques are evaluated on multiple contrastive learning methods (SimCLR v1, MoCo v2, SwAV) and for multiple downstream tasks (image classification, segmentation, and object detection). The experiments convincingly demonstrate that the accelerated approach works well when pre-training for a relatively small number of epochs.

My main criticism is that the performance values reported in the paper do not always match what is reported in the literature. Presumably this is because the amount of pre-training is less than what was done in previous work. For example, SwAV reports achieving 75.3 top-1 on ImageNet with linear classification, while Table 1 in the paper only reports 70.1. Although the acceleration approach works well early in training, it is not clear whether the same benefits play out when one aims to achieve performance on par with that reported in the literature. More (longer) experiments are required to support this claim.

One would expect that, eventually, the approximation techniques introduced to speed up training, may have a negative effect on the ultimate performance. It would be useful to know at what point in training (or at what scale) this becomes apparent. And if it is not the case that the approximation techniques cause any loss in performance, it would be nice to have some deeper understanding of why that's the case.

The claim that the proposed approach reduces the (average) memory usage during training is accurate, although it should be clarified that this is the _average_ memory usage. The peak memory usage still occurs when all layers are used to produce the representation, and this is no different than the peak memory usage of standard methods. Anyways, I guess the main message is about reduction in training time, and memory savings could be down-played.

Fine-tuning the full architecture generally performs much better than just linear evaluation for image classification. Did you also investigate the performance of methods using fine-tuning? Also, the test accuracy reported in Fig 3(b) is much lower than that reported in the SimCLRv1 paper (Table 6). Is it clear how Accel-simCLR behaves when run for more pre-training epochs?

Minor nit: Looking at the values in Table 1, I wonder why runtime isn't proportional to epochs for MoCo v2 (for both the standard and accelerated versions).

How many replicas are standard deviations computed over?

---

> ### Author Response · Authors · 2020-11-18
> **Reponse to Questions**
>
> We thank reviewer#1 for the thoughtful feedback!
>
> “My main criticism is that the performance values reported in the paper do not always match what is reported in the literature.”
>
> -- This is due to the different hyper-parameters. For example, SWAV achieves 75.3% top-1 accuracy when it uses multi-crop and large batch size. We have re-done the experiments on SwAV, and achieve similar performance with less training time compared to the 74.3% accuracy (256 batch size, 400 epochs, reported in Table 3 in SwAV paper).  Please refer to appendix B.3 in the revision for the new results.
>
> “It would be useful to know at what point in training (or at what scale) this becomes apparent.”
>
> -- This is a very good point. As shown in figure 3, our approach achieves the same performance as MOCO V2 when we train the model over 1000 epochs with our method. For SIMCLR, however, under the same training time, we still perform better. In practice, we notice the changing point depends on different hyper-parameters and different methods. We will explore this more.
>
> "The peak memory usage still occurs when all layers are used to produce the representation, and this is no different than the peak memory usage of standard methods."
>
> -- We agree that the random layerwise training can not save peak memory. However, we also proposed hard pair selection, which reduces the resolution of the feature maps in the intermediate stages, to cut the peak memory usage. We have clarified this in the experiment section, page 5.
>
> “Fine-tuning the full architecture generally performs much better than just linear evaluation for image classification. Did you also investigate the performance of methods using fine-tuning?”
>
> -- We have investigated finetuning in the paper; see Table 2 and 3 on page 7. Finetuning refers to the `not freeze backbone’ setting in these two tables. We can see that for these tasks, finetuning does achieve better performance as you expected. In all these settings, our approach does not have a performance drop.
>
> “Also, the test accuracy reported in Fig 3(b) is much lower than that reported in the SimCLRv1 paper (Table 6). Is it clear how Accel-simCLR behaves when run for more pre-training epochs?”
>
> -- This is mainly due to different hyper-parameters. The simCLR paper achieves ~62% accuracy when training the model with 200 epochs and 256 batch size (see Figure 9 of the simCLR paper).
> In comparison, we achieve comparable results (61.7%) when the batch size is 256. For the settings with larger batch size (e.g. 8192), we could not replicate the settings with larger batch size (e.g. 8192) due to the lack of computation resources for efficient parallel computing.
>
> “Looking at the values in Table 1, I wonder why runtime isn't proportional to epochs for MoCo v2 (for both the standard and accelerated versions).”
>
> -- Thanks for pointing out this. When reporting the results, we first generate a set (e.g. {1, 1.5, 2, ..}) by a scale (e.g. 0.5), and then round each number to a nearby number in the set. We approximate the training time by a scale of 0.5 in Table 1. In the revision, we report the results with a more precise scale (0.1 hours) in Table 1.
>
> "How many replicas are standard deviations computed over?"
>
> -- For detection and segmentation, as mentioned in the paper, the results are averaged over 5 trials. In the experiment, the linear classification results are also averaged over 5 trials.

---

> > ### Comment · AnonReviewer1 · 2020-11-24
> > **Acknowledging the authors response**
> >
> > Thank you for your response to the questions in my original review.
> >
> > In this case, the responses actually raise more questions and concerns, from my perspective.
> >
> > In a few responses you point out that differences in performance are due to using different hyperparameters. If performance is so sensitive to choice of hyperparameters I have some concerns that applying the approach proposed in this paper may not be practical, especially if finding good hyperparameter settings requires much additional effort.
> >
> > I also still have the concern that, when applied at scale (larger batch sizes), the results may no longer carry through in the same way.
> >
> > Thank you for pointing out that full fine-tuning is reported for the segmentation and detection tasks. Why not also report it for the classification task?
> >
> > Based on all of this, at the moment, I am inclined to reduce my score. I will take all of this into account during the discussion with the AC and reviewers, when determining my final score and recommendation.

---

> > > ### Author Response · Authors · 2020-11-25
> > > **Further clarification about the Background**
> > >
> > > Dear Reviewer1, thank you for your new comment.
> > > Here, we would like to clarify some background.
> > > 1) For the first problem (hyper-parameter), as shown in the appendix, the self-supervised learning methods that we built our algorithms upon (MoCo, SimCLR, SwAV) are sensitive to some hyper-parameters. (**For example, as shown in SimCLR and MoCo paper, their performance is sensitive to batch size and number of epochs.**) We show the results under different hyper-parameter settings (e.g. batch size, number of epochs, etc.) to demonstrate the robustness and generalizability of **our proposed method** under different settings. It is not proper to say **our proposed method** is sensitive to hyper-parameters. Indeed, as shown in the appendix, our method is not sensitive to the hyper-parameters introduced by us.
> > > 2) We do not report the fine-tuning results for classification. Actually, **all the previous works do not do this on ImageNet classification**. Recall that all the methods pre-train their backbone model on ImageNet dataset. Once you end-to-end fine-tune the pre-trained model on ImageNet classification, the model will end up with the same performance as supervised learning. In this case, the comparison will be meaningless because all the methods get nearly the same test accuracy. Hence, **people only report the linear classification result on ImageNet, with fixed pre-trained backbone**.
> > > For other downstream tasks, like object detection on PASCAL, the data domain is sufficiently far away from ImageNet, and different pre-trained models will have obvious difference w.r.t. performance after fine-tuning. Hence, we report both fixed and fine-tuning results **following previous works like MoCo**.

---

### Official Review · AnonReviewer3 · 2020-10-29
**Adding intermediate losses and hard pair selection to speed up the training of contrastive learning**

**Rating:** 6
**Confidence:** 3

**Review:**

**Summary**:
This paper proposes a new pipeline to speed up the training of contrastive learning. Specifically, besides the contrastive loss placed at the very end of the network, it introduces several additional intermediate losses. During the training, only one of them is used to compute gradients. With these intermediate losses, it can be also used to filter out the easy pairs. These two strategies can significantly accelerate contrastive learning while matching the performance of the recent methods. Authors conduct experiments and ablation studies to show the effectiveness of the proposed method on ImageNet and downstream tasks.

**Pros**:
+ The whole idea makes sense.  It introduces the intermediate losses to reduce the computation costs, and filter out easy samples to  recover the performance.
+ Authors conduct experiments on linear classification and downstream task transfer learning to show good performance.
+ Overall, the paper is well written and well organized.

**Concerns**:
- As we add multiple intermediate contrastive losses to the network, can we also evaluate the linear classification and downstream transfer task using the intermediate representations (after the project head)? I am curious about whether they can also obtain good performance.
- I wonder what if we update the network using all loss L, but not randomly sampling one of them? Will it improve the baseline performance?
- Does the hyper-parameter M affect the performance a lot when doing the hard pair selection? Can we do the ablation study on it?
- As mentioned in [1], the low- and mid-level representations, not high-level presentations, make the instance discrimination good for the detection task. This may also support the effectiveness of adding intermediate losses.

[1] What makes instance discrimination good for transfer learning?

**Minor Comments**:
* Page 2 (middle), below eq(2), same lass -> same class

Overall, I prefer the rating as above the threshold at the current stage. Hope the authors could address my concerns or questions in the rebuttal period.

---

> ### Author Response · Authors · 2020-11-18
> **Reponse to Questions**
>
> We thank reviewer#3 for the thoughtful feedback!
>
> "As we add multiple intermediate contrastive losses to the network, can we also evaluate the linear classification and downstream transfer task using the intermediate representations (after the project head)? I am curious about whether they can also obtain good performance."
>
> -- We have added the results of using intermediate representation for image classification in appendix [B.2] in the revision. For this task, we observe that using the intermediate loss can make the intermediate representation perform better than the intermediate representation of standard training. For more comparison, we refer to Appendix B.2 for details.
>
> "I wonder what if we update the network using all loss L, but not randomly sampling one of them? Will it improve the baseline performance?"
>
> -- We have tried this approach and found that it could not improve the performance over the randomly-sampling version.
>
> "Does the hyper-parameter M affect the performance a lot when doing the hard pair selection? Can we do the ablation study on it?"
>
> -- We observe that the hyper-parameter M does not affect the performance significantly. As shown in Appendix B.1, we achieve almost the same performance once M>=2. We have added more discussion in appendix B.1.

---

### Official Review · AnonReviewer2 · 2020-11-03

**Rating:** 5
**Confidence:** 5

**Review:**

Summary:

The paper proposes a clever trick to make instance contrastive learning faster by using the intermediate feature layers to perform the contrastive loss rather than just using the final 2048-d mean-pooled features as is typically done in MoCo or SimCLR. That way, the backprop costs are cheaper. The authors also use this intermediate representation based similarities to guide a better set of negatives to the layers on top. The authors demonstrate good results that show better time to accuracy (on the linear classifier) with both MoCo and SimCLR. A useful consquence of this paper is making contrastive unsupervised learning on ImageNet more accessible to people with less computation resources, ex PhD students in academic labs who may not have a DGX-1 or v3-TPU pods.

Pros:

Tackles an important problem - how can we make self-supervised learning even faster with clever engineering
Delivers on the problem by proposing two sensible solutions - saving the cost of backprop by using intermediate layers for contrastive losses, and also better negative mining while doing the fprop in a computationally efficient manner.
Good results in terms of time to accuracy on linear classifier - both with MoCo and SimCLR - two leading instance contrastive learning approaches.
Cons:

No code release as of yet. I believe the utility value of engineering driven papers is high but heavily relies of clean open source code that's usable by the community.
No results with fewer GPUs. If you can get results faster but still use 8 GPUs (though I do notice you have used 12 GB memory Titan V as opposed to 16 Gb memory GPUs used by MoCo, I don't think that's a deal breaker since MoCo ResNet-50s can be trained with the same HBM too). I think what's more important is to show you need fewer GPUs - can someone with a single GPU get results with the same amount of compute-time taken by Facebook to get MoCo's results.. That's what really increases accessibility to folks with less compute IMHO.
Reducing time to accuracy could be done in a host of other ways - eg - make everything run in fp16 or bfloat16 on the right kind of hardware - you would see a speedup of ~1.3x already. Use smaller image sizes while training, eg - instead of 224x224, use 192x192, use different temperatures (tune hyperparams) for training within as few epochs as possible. Use less unlabeled data to get the same accuracy. All these are orthogonal to your proposed method ofc, so not trying to compare them with yours. My point is that - why not push the broader goal to the extreme - can one combine all these engineering tricks in one piece, and design a way faster contrastive learning pipeline that can train with way less resources and still perform as well in much less wall clock time..
The Related work section can be improved - example, Contrastive Learning prior work should mention the CPCv2 work (Henaff et al 2019); and the idea of using intermediate layers and backprop. the losses through them should mention the Inception architecture (Szegedy et al. 2015) as well as Deep Classification work that did this few years ago for supervised learning.


Rating: Weak Reject - I would be open to reconsidering this if the authors commit to releasing their code, and also performing some benchmarking with respect to number of GPUs, and revising related work. I also encourage the authors to push further on making their pipeline even more efficient.

---

> ### Author Response · Authors · 2020-11-18
> **Reponse to Questions**
>
> We thank reviewer#2 for the feedback!
>
> “releasing code”
>
> -- We promise to release the code. Here is the link to our current implementation of simCLR:  [https://drive.google.com/drive/folders/1r0v4niEjotLsuVqbx9vw7qCKuCUms08w?usp=sharing]. The implementation is based on an open-source library OpenSelfSup. We will further clean the code and release the other parts.
>
> “related work”
>
> -- Thanks for your advice. We have added more discussion about these related works, including CPCv2, deep cluster and some other work for self-supervised learning, and Inception Net and DARTs for intermediate losses.
>
> “more ablation and extremely accelerating”
>
> -- Thanks for your advice. We view the extremely accelerating models as future work. We have already tried some of the approaches you mentioned, but have not found universal improvement. For example, we found that reducing image resolution from 224x224 to 192x192 during training hurts the performance of MOCO and MOCO V2. We show the results in appendix B.4. We will study more in future work.

---

### Decision · Program_Chairs · 2021-01-07
**Final Decision**

**Decision:**

Reject

**Comment:**

This paper introduces modifications that allow to make the training of contrastive-learning-based models practical. The goal of the paper is very interesting, and the motivation clear. This paper tackles a very important issue with recent unsupervised feature learning methods.
However, while the goal is great, the present submission does not provide time improvements on par with the ambitions of this work. As noted by R2, many other hacks could be used in conjunction with the current work to scale this goal to the extreme, yielding time improvements which would be of a more impressive magnitude. In its current form, this paper unfortunately doesn’t meet the bar of acceptance.
Given the interesting scope of this work, I strongly encourage the authors to take the feedback from reviews and discussions into account and submit to another venue.